# Intermolecular Interactions and In Vitro Performance of Methyl Anthranilate in Commercial Sunscreen Formulations

**Natércia d. N. Rodrigues *** , **Juan Cebrián, Anna Montané and Sandra Mendez**

Lipotec SAU, Calle Isaac Peral, 17 Pol. Ind. Camí Ral, 08850 Barcelona, Spain; juan.cebrian@lubrizol.com (J.C.); anna.montane@lubrizol.com (A.M.); sandra.mendez@lubrizol.com (S.M.)
* Correspondence: natercia.rodrigueslopes@lubrizol.com

**Abstract:** In order to afford the required level of broad-spectrum photoprotection against UV-B and UV-A radiation, sunscreens must contain a combination of UV filters. It is important that any interactions between UV filters do not adversely affect their photostability nor the overall photostability of the sunscreen formulation. In this work, we explore the feasibility of using methyl anthranilate (MA) as an alternative to the photo-unstable UV-A filter, avobenzone. From the in vitro studies presented here, we conclude that MA does not provide sufficient UV-A protection on its own but that it is more photostable in formulation than avobenzone. In addition, we found that both octocrylene (OCR) and ethylhexyl methoxycinnamate (EHMC), two commonly used UV-B filters, can stabilize MA through quenching of its triplet states, as previously reported, which has a demonstrable effect in formulation. In contrast with previously reported observations for mixtures of EHMC and avobenzone, we found no evidence of [2+2] photocycloadditions taking place between EHMC and MA. This work demonstrates how a clear insight into the photophysics and photochemistry of UV filters, as well as the interactions between them, can inform formulation design to predict sunscreen performance.

**Keywords:** sunscreen; spectroscopy; emulsion; photoprotection; photochemistry; photophysics; formulation; photostability; suncare; cosmetics; in vitro

## 1. Introduction

The increasing awareness of the damaging effects of excessive sun exposure [1] has led to a rise in demand for skincare products that provide protection against solar radiation, referred to as photoprotection. While the most effective means of photoprotection is to avoid direct sun exposure, by staying indoors or wearing protective clothing, sunscreens remain the public's preferred method to protect skin against photodamage [2]. Sunscreens are skincare products, usually in the form of lotions or sprays, that contain active ingredients which absorb, reflect, or otherwise block solar radiation before it can reach vulnerable skin cells [3].

When sunscreens first reached the market, their main purpose was to protect against photodamage caused by ultraviolet-B radiation (UV-B, 290–320 nm) [4,5]. UV-B radiation is directly absorbed by DNA and, as such, it leads to photoinduced DNA mutations that are often responsible for skin cancer [6]. However, there is growing evidence that UV-A radiation (320–400 nm) is also capable of generating DNA mutations [6]. In addition, UV-A has been shown to damage the skin through the generation of reactive oxygen species responsible for skin aging and oxidative stress that may indirectly lead to skin cancer [7,8]. UV-A radiation presents additional challenges compared to UV-B considering that it penetrates much deeper into the skin, it is approximately 20 times more abundant at the Earth's surface than UV-B, and it can be transmitted through glass, meaning skin can be exposed to UV-A even when indoors [9]. As such, modern suncare products contain a 'sun protection factor' (SPF), which relates mostly to UV-B protection, and also a 'UV-A protection factor' (UVAPF) [10]. The concept of 'critical wavelength,' defined as the wavelength at which 90% of the integrated UV absorbance is reached (from 290 nm to

400 nm), was also created to ensure broad spectrum photoprotection that sufficiently blocks UV-A radiation. Regulations require marketable sunscreens to have a critical wavelength of at least 370 nm [11,12].

Achieving appropriate UV-A protection remains a significant challenge to the suncare industry. These challenges are mainly related to the lack of strong and photostable UV-A absorbers that have regulatory approval for use in commercial sunscreens and that are available at a reasonable cost [13]. The most widely used UV-A filter on the market, avobenzone, is photo-unstable [14–16]. Upon absorption of UV-A radiation, avobenzone is known to undergo complex photochemistry, which includes tautomerization followed by several fragmentation pathways, taking place via the population of triplet states [17–19]. This photophysical and photochemical behavior is not ideal for an active sunscreen ingredient, also referred to as a UV filter, which should ideally dissipate energy quickly and efficiently in order to return to its initial state without generating photoproducts or inducing undesirable side chemistry [12]. Recent concerns regarding the percutaneous absorption of avobenzone have also placed it on a list of ingredients for which additional safety data are required, which may ultimately result in a change in regulation [20]. The current maximum permissible concentration of avobenzone (and other UV filters) in 'ready to use' formulations is given in Table 1.

**Table 1.** List of maximum permissible concentrations of UV filters used in the present experiments in 'ready to use' formulations, in the European Union (EU) and the United Stated of America (USA). Values are given in % *w/w*.

| UV Filter | Maximum Permissible Concentration (EU) * | Maximum Permissible Concentration (USA) ** |
|---|---|---|
| Ethylhexyl Triazone (Uvinul® T150) | 5% | Not approved as an UV filter |
| Bis-ethylhexylxyphenol Methoxyphenyl Triazine (Tinosorb® S) | 10% | Not approved as an UV filter |
| Ethylhexyl Methoxycinnamate (Neo Heliopan® AV) | 10% | 7.5% |
| Avobenzone (Butyl Methoxydibenzoylmethane, Parsol® 1789) | 5% | 3% |
| Octocrylene (Neo Heliopan® 303) | 10% | 10% |
| Methyl Anthranilate (CAS 134-20-3) | Not approved as an UV filter | Not approved as an UV filter |

* As per Regulation (EC) No 1223/2009 of the European Parliament and of the Council of 30 November 2009 on cosmetic products. ** As per the proposed rule 'Sunscreen Drug Products for Over-the-Counter Human Use' by the Food and Drug Administration on 26 February 2019.

The challenge of using avobenzone as a UV-A filter is further complicated by the fact that sunscreens always require a combination of UV filters in order to provide broad-spectrum photoprotection, which extends across UV-B and UV-A wavelengths. The stability of avobenzone is known to be affected when combined with other UV filters: for example, ethylhexyl methoxycinnamate (EHMC, Figure 1), can aggravate avobenzone's photoinstability, while octocrylene (OCR, Figure 1) stabilizes avobenzone [21–23]. The decreased photostability of sunscreen formulas containing a combination of avobenzone and EHMC is most likely due to the photoinduced [2+2] cycloaddition that takes place between these two filters, generating cyclobutene photoproducts [24]. The stabilizing effect of OCR on avobenzone is less clear but results previously reported by Lhiaubet-Vallet et al. suggest OCR acts as a quencher of avobenzone. In particular, OCR has been found to efficiently quench both the triplet states of avobenzone and the singlet oxygen they generate ($^1O_2$, a type of reactive oxygen species) while being itself relatively stable under these photosensitization conditions [25]. Nevertheless, the work by Lhiaubet-Vallet et al. also demonstrates that triplet states or $^1O_2$ quenching ability are not sufficient criteria to determine protection against photoinstability, since the systematic study of the effect of six different filters upon the photostability of avobenzone revealed no clear relationship between these parameters [25].

**Figure 1.** Molecular structures of MA (INCI name: methyl anthranilate), EHMC (INCI name: ethylhexyl methoxycinnamate) and OCR (INCI name: octocrylene). In addition, shown is the molecular structure of Meradimate, which consists of MA with an added menthyl unit, shown in blue.

In this work, we evaluate the suitability of methyl anthranilate (MA, Figure 1) as a direct replacement for avobenzone. MA is a precursor to Meradimate (INCI name: menthyl anthranilate), a UV-A filter which is not permitted for use in the European Union, Japan, nor Taiwan, despite still being approved for use up to 5% *w/w* in the United States, Australia, Brazil, Canada, South Africa, South Korea and the states belonging to the Association of Southeast Asian Nations (ASEAN) [26]. Unlike Meradimate, MA is widely approved for use in cosmetics as a fragrance [27,28], despite having been found to have the same spectroscopic, photophysical, and photochemical behavior of Meradimate [29]. In addition, we investigate the potential of MA for use as a UV-A filter by investigating its interactions with UV-B filters EHMC and OCR.

## 2. Materials and Methods

### 2.1. Sample Preparation

Formulations F1 to F6 were prepared in batches of 400 g following the formulas in Table 2 and according to the following procedure. First, the ingredients in the aqueous phase were added to water one by one under constant stirring with an overhead helix stirrer. The resulting mixture was left to disperse for approximately 1 h under stirring at approximately 700 rpm. In a separate beaker, the ingredients of the oil phase were mixed with a magnetic stirrer bar and heated to approximately 50 °C until a clear mixture was achieved. The temperature of the aqueous phase was also raised to 50 °C, and the hot oil phase was then added to the aqueous phase under constant stirring. The resulting mixture was homogenized using an Ultra-Turrax® disperser (IKA®-Werke GmbH & Co. KG, Staufen, Germany) rotating at 6000 rpm for 2 min. The mixture was then transferred back to mechanical stirring and left to cool to approximately 30 °C, after which sodium hydroxide (NaOH, 18% *w/w*) was added until a pH ~ 6.5 was reached. The preservative (Euxyl® PE 9010, Schülke and Mayr GmbH, Norderstedt, Germany) was added and mixed, and, finally, the formulation was transferred to a glass container for storage. Each batch was stored overnight in a chamber at 25 °C before further testing.

### 2.2. Photostability Tests

Square poly(methyl methacrylate) (PMMA) plates (25 cm$^2$), with a smooth surface on one side and textured on the other to mimic the skin surface, were first cleaned with deionized water. The plates were left to dry in a temperature-controlled chamber at 25 °C, and later transferred to a HD-Thermaster (HelioScreen, Creil, France), also set at 25 °C, ready for testing. Approximately 27.5 mg of each formulation were deposited onto the textured side of the PMMA plates in a square grid of 4 × 5 points. The formulations were then rubbed onto each plate with a gloved finger (nitrile gloves), first in rotative motions and then with swift back and forth motions between opposite edges of the square plate, in order to achieve an even distribution of the sample. The average sample coverage for these tests was approximately 0.3–0.4 mg/cm$^2$ in all cases; we note here that this was lower than the 2 mg/cm$^2$ recommended for sunscreen application [30] in order to avoid saturation of the detector, which would invalidate the results. After application, the sample coated

PMMA plates were allowed to rest in the closed (i.e., light-protected) HD-Thermaster at 25 °C for 30 min before testing. Blank plates were prepared in a similar fashion, using 15 mg of glycerin applied in a 3 × 3 grid.

**Table 2.** Breakdown of the ingredients in each of the formulas prepared and studied in the present work.

| Phase | Raw Material | Supplier, City, Country | Formula Number | | | | | |
|---|---|---|---|---|---|---|---|---|
| | | | F1 *w/w* % | F2 *w/w* % | F3 *w/w* % | F4 *w/w* % | F5 *w/w* % | F6 *w/w* % |
| Aqueous | Deionized water | - | 68.45 | 68.45 | 68.45 | 68.45 | 68.45 | 68.45 |
| | Disodium EDTA | Ricardo Molina S.A.U., Barcelona, Spain | 0.10 | 0.10 | 0.10 | 0.10 | 0.10 | 0.10 |
| | PEMULEN™ * EZ-4U | Lubrizol Advanced Materials, Barcelona, Spain | 0.15 | 0.15 | 0.15 | 0.15 | 0.15 | 0.15 |
| | CARBOPOL® * ULTREZ 30 | Lubrizol Advanced Materials, Barcelona, Spain | 0.20 | 0.20 | 0.20 | 0.20 | 0.20 | 0.20 |
| | GLUCAM™ * E-20 | Lubrizol Advanced Materials, Barcelona, Spain | 3.00 | 3.00 | 3.00 | 3.00 | 3.00 | 3.00 |
| Oil | GLUCAMATE™ * SSE-20 | Lubrizol Advanced Materials, Barcelona, Spain | 0.80 | 0.80 | 0.80 | 0.80 | 0.80 | 0.80 |
| | Ethylhexyl Triazone (Uvinul® T150) | BASF Española S. L., Barcelona, Spain | 1.00 | 1.00 | - | - | - | - |
| | Bis-ethylhexylxyphenol Methoxyphenyl Triazine (Tinosorb® S) | BASF Europe GmbH, Berlin, Germany | 2.50 | 2.50 | - | - | - | - |
| | Ethylhexyl Methoxycinnamate (Neo Heliopan® AV) | Symrise, Inc., Rennes, France | 4.00 | 4.00 | - | - | 9.70 | - |
| | Avobenzone (Butyl Methoxydibenzoylmethane) (Parsol® 1789) | DSM Nutritional Products Ltd., Heerlen, the Netherlands | 4.00 | - | - | - | - | - |
| | SCHERCEMOL™ * LL | Lubrizol Advanced Materials, Barcelona, Spain | 7.00 | 7.00 | 7.00 | 22.5 | 7.00 | 7.00 |
| | Octocrylene (Neo Heliopan® 303) | Symrise, Inc., Rennes, France | 8.00 | 8.00 | - | - | - | 9.70 |
| | Methyl Anthranilate (CAS 134-20-3) | Merck, Sigma-Adlrich, Madrid, Spain | - | 4.00 | 19.50 | 4.00 | 9.80 | 9.80 |
| Other | Sodium Hydroxide (NaOH, 18%) | - | 0.30 | 0.30 | 0.30 | 0.30 | 0.30 | 0.30 |
| | Phenoxyethanol and ethylhexylglycerin (Euxyl® PE 9010) | Schülke and Mayr GmbH, Norderstedt, Germany | 0.50 | 0.50 | 0.50 | 0.50 | 0.50 | 0.50 |

\* Trademark owned by The Lubrizol Corporation or its affiliates.

Each plate was analyzed before and after irradiation with a SUNTEST CPS + (III) solar simulator (Atlas Material Testing Solutions, Mount Prospect, IL, USA). This solar simulator delivers 550 W/m$^2$ over the 300–800 nm wavelength range, and the irradiation was carried out over 20 min, equating to a radiation dose of 660 kJ/m$^2$. The SPF and UVAPF values for each plate were obtained with a LabSphere UV-2000 ultraviolet (UV) transmittance analyzer (LabSphere, Inc., North Sutton, NH, USA), which calculates these values based on an average of readings from 5 different points on the plate. The final values presented in Table 3 are an average of 3 plates for each formulation.

Transmittance spectra for formulations F1 to F6 were also obtained with the LabSphere UV-2000 ultraviolet (UV) transmittance analyzer (LabSphere, Inc., North Sutton, NH, USA). Transmittance (T) spectra were converted into absorbance (A) spectra using the mathematical relationship A = log (1/T). The integration of each transmittance and absorbance curve was used as the indicator of protection afforded by each formulation. The percentage change (% change) in the integrated transmittance was calculated using

$$\% \text{ change} = \frac{T_{\text{after}} - T_{\text{before}}}{T_{\text{before}}} \times 100, \tag{1}$$

where $T_{\text{before}}$ and $T_{\text{after}}$ correspond to integrated transmittance before and after irradiation, respectively. The % changes in absorbance were calculated following the same rationale; these values are all presented in Table 4.

**Table 3.** Sun protection factor (SPF), UV-A protection factor (UVAPF), and critical wavelength values measured before and after irradiation of formulations F1 to F6. Errors are reported to one standard deviation.

| Formula | SPF | | UVAPF | | Critical Wavelength/nm | |
|---|---|---|---|---|---|---|
| | Before Irradiation | After Irradiation | Before Irradiation | After Irradiation | Before Irradiation | After Irradiation |
| F1 | 48 ± 1 | 43 ± 1 | 23.4 ± 0.1 | 21 ± 1 | 379.4 ± 0.3 | 378.8 ± 0.3 |
| F2 | 31 ± 2 | 25 ± 1 | 6.7 ± 0.2 | 5.9 ± 0.1 | 367.5 ± 0.4 | 368 ± 0 |
| F3 | 12 ± 1 | 4.8 ± 0.5 | 5.3 ± 0.2 | 3.7 ± 0.2 | 360 ± 0 | 360.5 ± 0.1 |
| F4 | 2.01 ± 0.04 | 1.48 ± 0.01 | 2.09 ± 0.04 | 1.52 ± 0.01 | 361.7 ± 0.1 | 370.7 ± 0.2 |
| F5 | 22 ± 1 | 11.6 ± 0.3 | 3.05 ± 0.03 | 2.47 ± 0.05 | 354.9 ± 0.1 | 355.5 ± 0.4 |
| F6 | 14 ± 1 | 11.3 ± 0.5 | 3.69 ± 0.05 | 2.81 ± 0.04 | 358.7 ± 0.2 | 356.3 ± 0.5 |

**Table 4.** Comparison in percentage (%) change in transmittance and absorbance observed for formulas F1 to F6. These percentages are calculated for integrated transmittance and absorbance, i.e., for the area under the transmittance and absorbance curves.

| Formula | % Change in Transmittance | % Change in Absorbance |
|---|---|---|
| F1 | 1.88 | −3.44 |
| F2 | 1.79 | −6.77 |
| F3 | 5.92 | −36.12 |
| F4 | 9.50 | −38.37 |
| F5 | 5.70 | −24.86 |
| F6 | 5.55 | −15.79 |

## 3. Results and Discussion

The first aim of this work was to evaluate if MA could be a suitable UV-A filter to substitute avobenzone in commercial sunscreen formulations. With this in mind, a reference sunscreen formulation containing avobenzone (formulation F1) was compared to formulation F2, for which avobenzone was directly replaced with MA (with no other alterations to the formula). The in vitro performance of formulation F2, as reported in Table 3, suggests that MA cannot serve as a direct substitute of avobenzone in a commercial photoprotective formulation: not only is the UVAPF significantly lower for the formulation prepared with MA (367.5 ± 0.4 before irradiation, compared to 379.4 ± 0.3 in F1), but its critical wavelength is also below the regulatory minimum required for a commercial suncare lotion (≥370 nm). The lower UVAPF, which also results in a lower overall SPF, is likely a direct result of the lower extraction coefficient of MA over the UV-A range, when compared to avobenzone. Nevertheless, MA could serve as an important UV-A protection booster in formulations with a combination of alternative UV-A filters, which may prove particularly important if regulation changes place further restrictions on the use of avobenzone in the future [20]. It is also entirely possible that, as is the case with avobenzone, the performance of MA might be improved by combination with UV-B filters, and as such, its photochemical behavior remains of interest.

One of the main points of interest in evaluating sunscreen performance is photostability, i.e., a comparison between the photoprotection afforded by the formulation before and after irradiation to ensure that the formulation can provide long-term photoprotection, with no reduction in performance upon exposure to solar radiation. Figure 2 shows transmittance and absorbance curves for formulations F1 and F2 before and after irradiation. Whilst both formulations can be considered photostable, the results in Figure 2 suggest that F2 (containing MA instead of avobenzone) is less photostable, which is confirmed by a starker drop in absorbance after irradiation for F2 when compared to F1 (see Table 4). The lack of photostability of MA is also confirmed by the spectra for formulations F3 and F4, as shown in Figure 3. Formulations F3 and F4 contain only MA and, therefore, demonstrate the intrinsic photostability of MA when free from interaction with other UV filters. It is

clear from visual inspection of the spectra in Figure 3 that MA-containing formulations experience a significant loss of photoprotection, with a nearly 40% drop in absorbance after irradiation (see Table 4). In a previous study, Afonso *et al.* compared the area under the absorbance curve (AUC) of an avobenzone-only sunscreen formulation before and after irradiation, having determined a $AUC_{after}/AUC_{before}$ ratio of approximately 0.4 [31]. The equivalent value for formulations F3 and F4 in the present work was, in both cases, ~0.6, revealing that despite the marked photodegradation of MA this UV-A filter is more photostable than avobenzone.

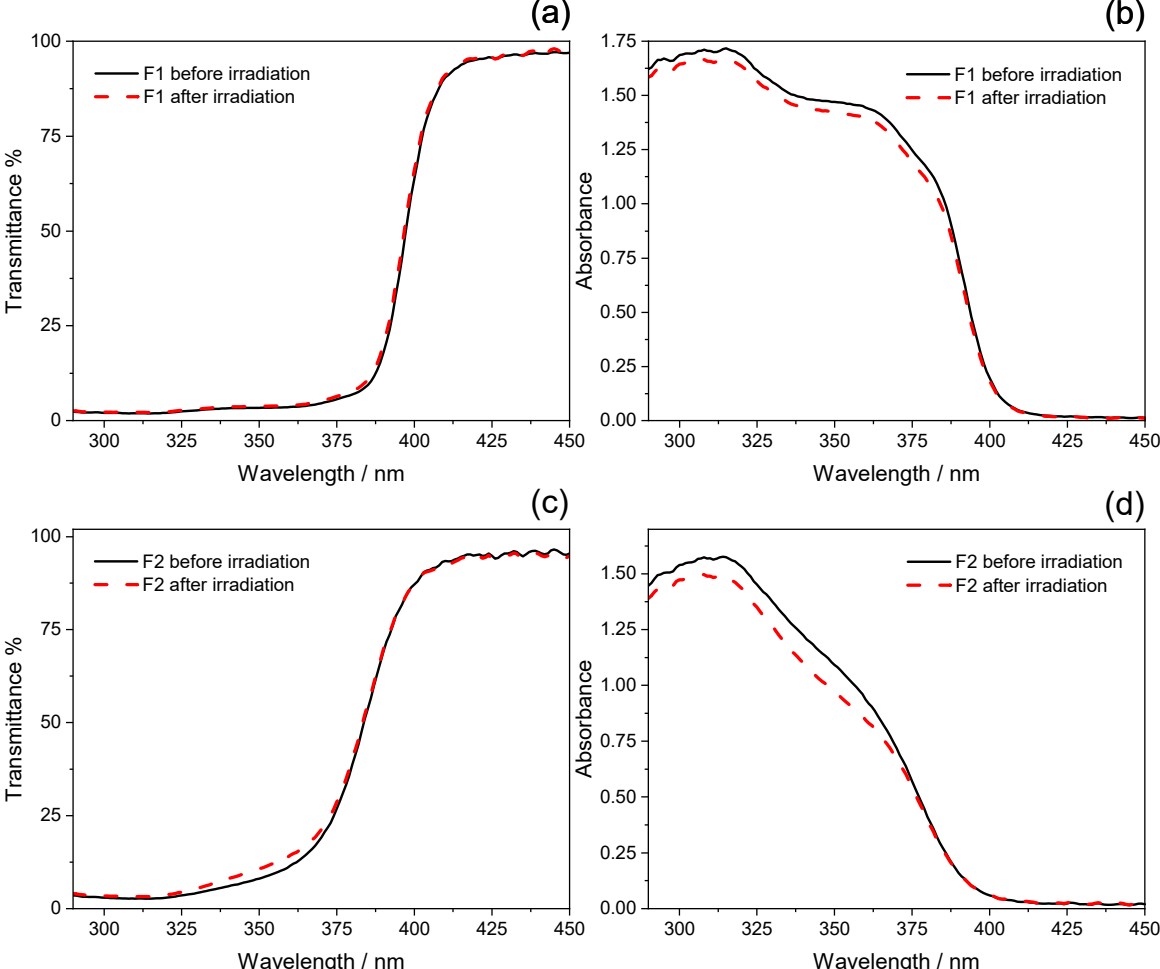

**Figure 2.** Transmittance (**a**,**c**) and absorbance (**b**,**d**) spectra obtained for formulations F1 (top) and F2 (bottom), before irradiation (solid black lines) and after irradiation (dashed red lines).

It is important to note here that the origin of the photo-instability observed in MA is different from that of avobenzone: while avobenzone is known to undergo molecular degradation, which generates photoproducts via population of triplet states [17–19], the intramolecular N–H–O bond in MA confers it some photostability, avoiding immediate bond breaking [29]. However, it was previously reported by Rodrigues et al. [29] that the excited state of MA accessed with UV-A irradiation has no accessible energy dissipation pathways. As such, the excess energy absorbed upon photoexcitation remains trapped in long-lived excited states, including triplet states, which may facilitate the side chemistry responsible for the observed loss of absorbance after irradiation. The existence of these long-lived states was evidenced by the relatively high yields of both fluorescence and phosphorescence measured for MA [29]; low fluorescence and phosphorescence quantum yields may, therefore, constitute a rapid and straightforward selection criterion for potential UV filter candidates in future development.

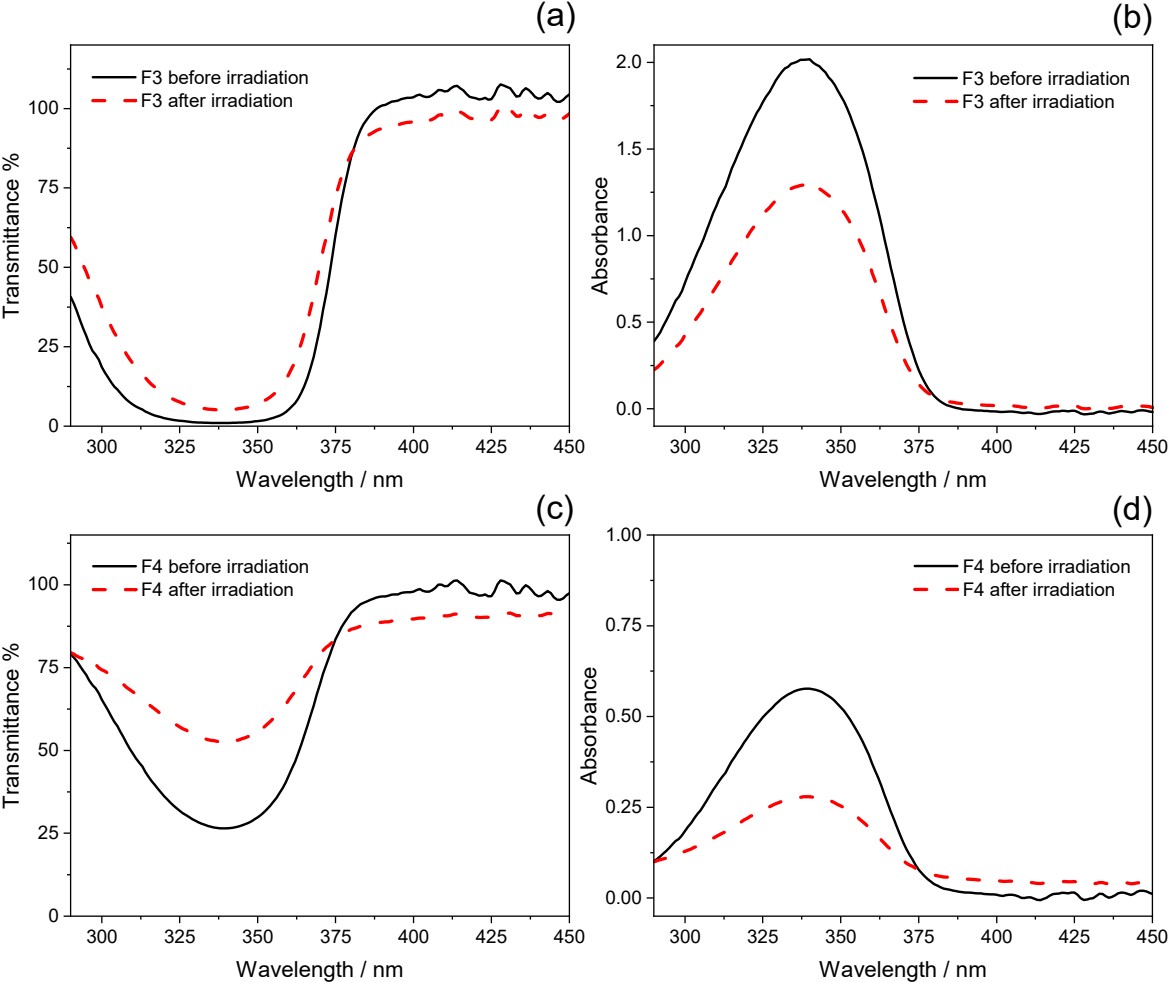

**Figure 3.** Transmittance (**a**,**c**) and absorbance (**b**,**d**) spectra obtained for formulations F3 (top) and F4 (bottom), before irradiation (solid black lines) and after irradiation (dashed red lines).

While formulations F1 and F2 contain several UV filters that will be interacting with each other, complicating the interpretation of the results obtained, the only difference between formulations F3 and F4 is the concentration of MA. Namely, formulation F3 has a much higher concentration of MA when compared to F4. Visual inspection of Figure 3a,c also suggests that formulation F3 is more photostable than formulation F4, given the more obvious increase in transmittance after irradiation of formulation F4. While this observation seems to be in disagreement with the relative drop in SPF for formulations F3 and F4 (see Table 3), the comparisons in Table 4 corroborate the increased photostability of F3. Since the SPF values in Table 3 are more vulnerable to propagated error, they are considered to be less reliable for the purpose of comparing relative changes before and after irradiation. Assuming the higher concentration of MA does indeed lead to a more photostable formulation (F3, in this case), this could be explained by a self-quenching mechanism: a process by which interaction between MA molecules themselves would quench excess energy and thus stabilize the mixture. This type of self-quenching mechanism has been observed in Meradimate, a close analog of MA with nearly identical photodynamics [29], even though it was noted to be a slow process in this case, with a rate constant of $9 \times 10^6$ dm$^3$ mol$^{-1}$ s$^{-1}$ [32]. Nevertheless, in a high concentration mixture such as formulation F3, intermolecular interactions are facilitated by the closer proximity between molecules, and self-quenching mechanisms are enhanced. It is therefore plausible to assume that self-quenching interactions would take place between MA molecules upon irradiation of these formulations.

In addition to self-quenching mechanisms, Meradimate has also been found to undergo diffusion-limited triplet-triplet energy transfer with other UV filters, namely EHMC

and OCR (molecular structures in Figure 1) [33]. In order to evaluate how these interactions affect formulation photostability, we prepared formulations F5 and F6, for which MA was combined with EHMC and OCR, respectively. Comparing the spectra obtained before and after irradiation of formulations F5 and F6 (see Figure 4) with previously presented results in Figure 3, it is clear that combining MA with either EHMC or OCR improves the photostability of MA. This is contrary to observations for avobenzone, for which EHMC has a destabilizing effect, as previously mentioned. In order to understand this discrepancy, a fundamental understanding of the mechanisms regulating the intermolecular interactions between MA and EHMC or OCR is necessary. Both EHMC and OCR act as triplet state quenchers when interacting with Meradimate [33] and, given the virtually identical photo-dynamics of Meradimate compared to MA [29], it is plausible to assume that EHMC and OCR would have similar quenching effects on MA. The significant enhancement in photo-stability achieved for MA when combined with OCR (F6) is most likely associated with this triplet quenching mechanism, by which OCR removes excess energy from MA, thus avoiding its photodegradation or energy transfer to other more susceptible components of the formulation.

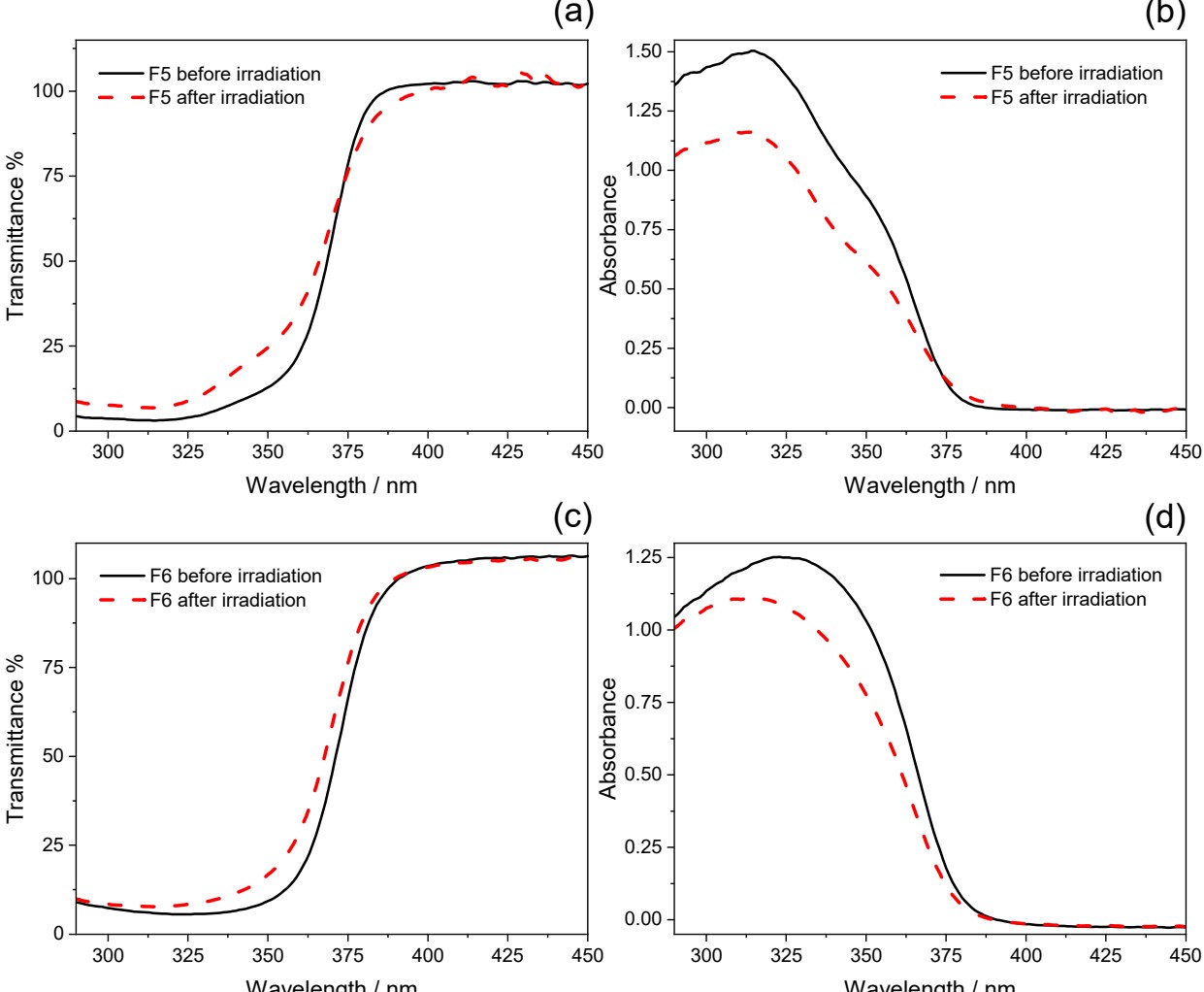

**Figure 4.** Transmittance (**a**,**c**) and absorbance (**b**,**d**) spectra obtained for formulations F5 (top) and F6 (bottom), before irradiation (solid black lines) and after irradiation (dashed red lines).

In the case of combining MA with EHMC (F5), the increase in photostability of MA is not as pronounced. This may be due to the fact that, unlike OCR, EHMC is itself not entirely photostable: EHMC has been widely reported to undergo *trans-cis* isomerization

upon irradiation leading to decreased absorbance [34,35]. However, there are two other photochemical processes known to occur in photoexcited EHMC that warrant further investigation. Firstly, it has been reported that, at high concentrations, EHMC undergoes a [2+2] photocycloaddition reaction mechanism resulting in 13 possible different structures of dimers. At the high concentration of EHMC used in these formulations, it is plausible that these [2+2] photocycloaddition processes would become competitive over energy transfer mechanisms, leading to loss of EHMC and, therefore, to loss of absorbance after irradiation. In addition, EHMC could potentially undergo [2+2] photocycloaddition reactions with MA, similar to its interaction with avobenzone, as previously described [24].

In order to ascertain whether EHMC does undergo these [2+2] photocycloaddition reactions in the formulations under study, we analyzed three mixtures containing SCHERCEMOL™ LL as the emollient and either (A) MA, (B) EHMC, or (C) a mixture of MA and EHMC, with concentrations comparable to those used in the oil phases of the formulas presented in Table 2. These mixtures were irradiated for two hours with a solar simulator and then analyzed by high-performance liquid chromatography or mass spectrometry (see Supplementary Materials). While MA has previously been found to undergo photodegradation with a quantum yield of 0.17% under simulated solar light [36], we were unable to unambiguously identify any such photoproducts in the irradiated mixture A. In addition, irradiation of MA with 300 nm light has previously been reported to initiate the formation of MA trimers [37], but once again, we found no evidence of these species in our studies. In the case of mixture B, containing only EHMC and SCHERCEMOL™ LL, we have found some evidence of its *cis* isomer [34] or of a dimer, as suggested by Herzog et al. [35] (see Supplementary Materials). Importantly, no other photoproducts are detected in mixture C, which contained a combination of MA and EHMC, suggesting that [2+2] photocycloaddition reactions do not take place between these two molecules unlike the case of EHMC and avobenzone [24]. As such, the decreased effect of EHMC in improving the photostability of the MA-containing formulation (F5) is not due to any photochemistry taking place between MA and EHMC, but more likely due to the photochemistry of EHMC (*trans-cis* photoisomerization and dimerization, as previously mentioned).

This work demonstrates how a fundamental understanding of the complex photophysics and photochemistry of UV filters is crucial to understanding sunscreen formulation performance, particularly regarding photostability. For example, although photoexcited MA creates triplet states [29], it is not sufficient to assume that a triplet state quencher will stabilize it; avobenzone also undergoes photochemistry via triplet states but is not stabilized by EHMC, which functions as a triplet quencher for MA. Similarly, while triplet states often lead to the generation of reactive oxygen species, particularly singlet oxygen, it may not be necessarily true that a singlet oxygen quencher will improve the photostability of the components of a formulation. Ultimately, a combination of energy transfer mechanisms determines what chemistry takes place upon irradiation of a sunscreen formulation. Therefore, a fundamental understanding of the rich photophysics and photochemistry that occur in photoexcited UV filters, and, in particular, in mixtures of UV filters, allows for smart design of sunscreen formulation by enabling the formulator to better predict sunscreen performance.

## 4. Conclusions

In this work, we have determined that MA does not provide sufficient photoprotection to justify its use in sunscreen formulations as a direct alternative to photo-unstable avobenzone. MA could, nevertheless, be used as a UVAPF booster in combination with other filters. We have found that both EHMC and OCR improve the photostability of MA in formulation, likely via triplet state quenching. This is in contrast with the case of avobenzone, for which EHMC has a destabilizing effect. Importantly, EHMC does not undergo [2+2] photocycloadditions with MA, as it does with avobenzone, justifying the stabilizing effect in this case.

The work presented highlights the crucial role that a fundamental understanding of UV filters plays in the smart formulation of sunscreens and other suncare products. Specifically, it demonstrates the type of rationale that can guide a strategic approach to enhancing sunscreen photostability and performance: gathering a thorough understanding of the relaxation pathways of each UV filter, how these are affected by combination with other filters and, importantly, mapping the fate of excess energy as it is transferred between filters to ensure its safe dissipation. This insight can inform a smart formulation that is better able to predict not only the photostability of a sunscreen formulation but also its potential to generate harmful reactive species on the skin upon sun exposure.

**Supplementary Materials:** The following are available online at https://www.mdpi.com/article/10.3390/appliedchem1010005/s1, Figure S1: chromatogram obtained for mixture A (containing only MA) at 220 nm detection, before and after irradiation; Figure S2: chromatogram obtained for mixture A (containing only MA) at 300 nm detection, before and after irradiation; Figure S3: chromatogram obtained for mixture B (containing only EHMC) at 220 nm detection, before and after irradiation; Figure S4: chromatogram obtained for mixture B (containing only EHMC) at 300 nm detection, before and after irradiation; Figure S5: chromatogram obtained for mixture C (containing MA and EHMC) at 220 nm detection, before and after irradiation; Figure S6: chromatogram obtained for mixture C (containing MA and EHMC) at 300 nm detection, before and after irradiation; Figure S7: mass spectrum obtained for mixture A (containing only MA) post-irradiation; Figure S8: mass spectrum obtained for mixture B (containing only EHMC) post-irradiation; Figure S9: mass spectrum obtained for mixture C (containing MA and EHMC) post-irradiation.

**Author Contributions:** N.d.N.R. was responsible for conceptualization, methodology, validation, formal analysis, investigation, resources, data curation, writing, review, editing, project administration and funding acquisition. J.C. provided guidance and supervised this work. A.M. was responsible for investigation. S.M. was responsible for supervision of the work carried out by A.M. All authors have read and agreed to the published version of the manuscript.

**Funding:** This research was funded by the European Union's Horizon 2020 research and innovation program under grant agreement number 844177–SUNNRL.

**Institutional Review Board Statement:** Not applicable.

**Informed Consent Statement:** Not applicable.

**Data Availability Statement:** Any data not presented in this manuscript can be found in the Supplementary Materials (see above).

**Acknowledgments:** The authors would like to thank Mireia Martín and Pedro Guardeno for helpful discussion and guidance in formulation strategies. N.d.N.R. further acknowledges the support and funding received from the European Union's Horizon 2020 research and innovation program under the Marie Skłodowska-Curie grant agreement No 844177–SUNNRL.

**Conflicts of Interest:** All authors are employees of The Lubrizol Corporation (Lipotec, Barcelona, Spain).

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
