# Peer review of "Intermolecular Interactions and In Vitro Performance of Methyl Anthranilate in Commercial Sunscreen Formulations"

_appliedchem, doi:10.3390/appliedchem1010005_

Round 1

Reviewer 1 Report

The paper by Rodrigues et al. reports a study on the performance of methyl anthranilate (MA) as potential sunscreens in commercial formulations. The authors have explored six different formulations including MA alone or in combination with other well-known commercial sunscreens to determine the utility of this additive to provide photoprotection.

The methodology used in the paper is a standard in the field. The experiments are meaningful and well described. The presentation is good and the paper is well written. Thus, all the formal aspects are more than adequate.

The overall feeling of this paper is quite positive. However, there are some issues that, in my opinion, would greatly increase the quality and potential impact of the manuscript. The narrative of the authors moves between the observational nature of the results and the speculative presentation of conclusions. While the experimental design and observations are clear and illustrative, in some cases the authors try to get deeper into the photophysical behaviour of the mechanisms involved without proper data. As a consequence, the conclusions drawn by the authors (even if potentially correct) are not supported by data and render the paper too speculative and fatuous.

This paper has clearly some merits and deserves publication. However, I would suggest the authors stick to the analysis of the data, without trying to go beyond with risky conclusions, or, even better, to try to deepen more in the photophysics of MA.

Some specific examples of assertions not fully supported by data:

-6: "However, the excited state of MA that is accessed with UV-A irradiation has no accessible energy dissipation pathways." 

This would require an extended study and cannot be understood from the present data.

-p6: "Formulation F3 has a much higher concentration of MA and seems to be more photostable than formulation F4"

This is not clearly shown in Table 2. The decrease in SPF for F3 is 60% while for F4 the decrease is just 26%

-p6: "This concentration dependent effect suggests a self-quenching mechanism: a process by which interaction between MA molecules themselves can quench excess energy and thus stabilize the mixture"

This should be tested by a specially designed experiment with different concentration values, not just by two formulations.

In addition, the quenching effect will be probably hampered by viscosity in the formulation. This should be checked.

-p7: "Both EHMC and OCR act as triplet state quenchers when interacting with Meradimate [33] and it can be assumed they would have a similar effect on MA."

This assumption is not trivial. Even if they could act as quenchers, this should be checked. In fact, the authors discuss the different effects of quenching when it comes to different sunscreens, such as avobenzone.

-p8-9: The discussion about photochemistry (Figure 5) based at least partially on the colour of the samples is extremely speculative

Author Response

We thank the reviewer for their time in reading and evaluating this manuscript. We are furthermore thankful for both the kind words regarding the manuscript’s quality, and for the suggestions made, which we agree will further improve the quality of the paper. We have attempted to address the reviewer’s comments to the best of our ability; changes made to the manuscript as a response to these comments are highlighted in the reviewed manuscript in blue. We also take each of Reviewer 1’s comments and address them individually here:

"However, the excited state of MA that is accessed with UV-A irradiation has no accessible energy dissipation pathways." This would require an extended study and cannot be understood from the present data.

We agree that this statement is unsupported by any evidence that was presented in the paper. The statement refers, however, to previously reported work, which has shown this to be the case. In order to clarify this point, we have added wording to this effect and re-emphasized the corresponding reference. The statement now reads: “However, as has previously been reported by Rodrigues et al.,[29] the excited state of MA that is accessed with UV-A irradiation has no accessible energy dissipation pathways.

"Formulation F3 has a much higher concentration of MA and seems to be more photostable than formulation F4". This is not clearly shown in Table 2. The decrease in SPF for F3 is 60% while for F4 the decrease is just 26%.

We would argue here that the values presented in Table 3 (now Table 4, following changes to the manuscript following comments by Reviewer 2), which refer to the areas under the transmittance and absorbance curves of each sample, are a better indicator of photostability than relative drops is SPF. It is an empirical observation that in vitro SPF measurements tend to have larger errors for formulas with higher UV filter concentrations which, in addition to errors associated with the SPF measurements, could render relative variations in SPF (before and after irradiation) less reliable. In addition, visual inspection of the transmittance curves in Figure 3 (a) and (c) in the main manuscript strongly suggest that formulation F4 is indeed less photostable than formulation F3.

However, we accept that this conclusion is not unambiguous, nor are the observations as conclusive as the writing has suggested. To reflect this, we have slightly altered the discussion surrounding the sentence highlighted by the reviewer, and it now reads, in lines 213-226:

While formulations F1 and F2 contain several UV filters that will be interacting with each other, complicating the interpretation of the results obtained, the only difference between formulations F3 and F4 is the concentration of MA. Namely, formulation F3 has a much higher concentration of MA when compared to F4. Visual inspection of Figure 3 (a) and (c) suggests also that formulation F3 is more photostable than formulation F4, given the more obvious increase in transmittance after irradiation of formulation F4. While this observation seems to be in disagreement with the relative drop in SPF for formulations F3 and F4 (see Table 3), the comparisons in Table 4 corroborate the increased photostability of F3. Since the SPF values in Table 3 are more vulnerable to propagated error, they are considered to be less reliable for the purpose of comparing relative changes before and after irradiation. Assuming the higher concentration of MA does indeed lead to a more photostable formulation (F3, in this case), this could be explained by a self-quenching mechanism: a process by which interaction between MA molecules themselves would quench excess energy and thus stabilize the mixture.

"This concentration dependent effect suggests a self-quenching mechanism: a process by which interaction between MA molecules themselves can quench excess energy and thus stabilize the mixture". This should be tested by a specially designed experiment with different concentration values, not just by two formulations. In addition, the quenching effect will be probably hampered by viscosity in the formulation. This should be checked.

We agree with the reviewer that our suggestion of a self-quenching mechanisms would need further experimental validation, with experiments that are unfortunately not accessible to us in our facilities. Nevertheless, the suggestion of self-quenching for MA is based on comparison to its close analogue, Meradimate. While we recognize that it cannot be assumed that two analogues will display the same spectroscopic and/or energy transfer behaviors, we support this comparison with previously reported studies which have shown that MA and Meradimate have nearly identical photodynamics (Rodrigues et al., reference 29 in the main manuscript). To emphasize this similarity that draws us to suggest self-quenching mechanisms in MA, we have slightly re-worded the text in lines 226-229 in the main manuscript, which now reads: “This type of self-quenching mechanism has been observed in Meradimate, a close analogue of MA with nearly identical photodynamics,[29] even though it was noted to be a slow process in this case, with a rate constant of (…).

Furthermore, we agree with the reviewer’s comment regarding the effect of viscosity in this potential self-quenching mechanism. The effects of formulation viscosity on UV filter photodynamics is beyond the scope of the current study but it is, however, something we will endeavor to explore in the near future.

"Both EHMC and OCR act as triplet state quenchers when interacting with Meradimate [33] and it can be assumed they would have a similar effect on MA." This assumption is not trivial. Even if they could act as quenchers, this should be checked. In fact, the authors discuss the different effects of quenching when it comes to different sunscreens, such as avobenzone.

We agree with the reviewer that it would not be correct to assume that a molecule that acts as a quencher of species A would also quench species B. In fact, such an assumption is not valid, as we go on to argue with respect to the effects of EHMC on avobenzone, which are contrary to its effects on MA.

However, as just discussed in the previous point, the assumption made here is based on the fact that the spectroscopic behavior of MA has been found to be virtually identical to that of Meradimate, down to the fine details of its ultrafast photodynamics, as previously reported by Rodrigues et al., reference 29 in the main manuscript. It is because MA and Meradimate are so similar in their photodynamics that we suggest here that the effects of quenchers such as EHMC and OCR would also be similar in both MA and Meradimate. In addition, the data presented in this work also seems to suggest that EHMC does indeed stabilize MA, which is an indicator of this quenching. We acknowledge, however, that this would of course need to be confirmed and validated by studies similar to those undertaken by Matsumoto et al. for Meradimate, as reported on in reference 33 in the main manuscript. Such a study is, however, both outside the scope of this work and beyond the experimental facilities to which we have access.

Nevertheless, in acknowledgement that the sentence highlighted here by the reviewer does not clearly reflect the considerations above, we have reworded it so that it now reads: “Both EHMC and OCR act as triplet state quenchers when interacting with Meradimate [33] and, given the virtually identical photodynamics of Meradimate compared to MA,[29] it is plausible to assume that EHMC and OCR would have similar quenching effects on MA.

The discussion about photochemistry (Figure 5) based at least partially on the colour of the samples is extremely speculative

Also in response to comments from Reviewer 2, we have removed Figure 5 from the manuscript. In addition, in order to make the discussion surrounding our HPLC and mass spectrometry results more concrete, we have re-worded this segment of the paper, which now reads:

"In order to ascertain whether EHMC does undergo these [2+2] photocycloaddition reactions in the formulations under study, we analyzed three mixtures containing SCHERCEMOLTM LL as the emollient and either (A) MA, (B) EHMC or (C) a mixture of MA and EHMC, with concentrations comparable to those used in the oil phases of the formulas presented in Table 1. These mixtures were irradiated for two hours with a solar simulator and then analyzed by high performance liquid chromatography or mass spectrometry (see Supplementary Materials). While MA has previously been found to undergo photodegradation with a quantum yield of 0.17% under simulated solar light,[36] we were unable to unambiguously identify any such photoproducts in the irradiated mixture A. In addition, irradiation of MA with 300 nm light has previously been reported to initiate formation of MA trimers,[37] but once again we found no evidence of these species in our studies. In the case of mixture B, containing only EHMC and SCHERCEMOLTM LL, we have found some evidence its cis isomer [34] or of a dimer, as suggested by Herzog et al.[35] (see Supplementary Materials). Importantly, no other photoproducts are detected in mixture C, which contained a combination of MA and EHMC, suggesting that [2+2] photocycloaddition reactions do not take place between these two molecules, unlike the case of EHMC and avobenzone.[24] As such, the decreased effect of EHMC in improving the photostability of the MA-containing formulation (F5) is not due to any photochemistry taking place between MA and EHMC, but more likely due to the photochemistry of EHMC (trans-cis photoisomerization and dimerization, as previously mentioned)."

 We hope these changes are acceptable to the reviewer, making the manuscript fit for publication.

Reviewer 2 Report

  1. Υοu have to underline strictly the chemical difference between methyl anthranilate and meradimate. I do not understand (lines 82-89). Please make it clear by chemical structures.
  2. The permitted concentrations in Europe and USA of the used  UV filters in the presented experiments have to be referred in a table.
  3. Figure 5 is not necessary. The are UV spectrums (in vitro SPF, UVAPF) in the manuscript. Photographs of the irradiated solutions do not improve the quality of the manuscript
  4. More precision in the title, abstact and key words (the experiments are in vitro and that has to be referred, as well)

Author Response

We thank the reviewer for their time in reading and evaluating this manuscript, and for the comments which we understand will improve the quality of the paper. We have attempted to address the reviewer’s comments to the best of our ability; changes made to the manuscript as a response to these comments are highlighted in the reviewed manuscript in red.

We also take each of Reviewer 2’s comments and address them here individually:

With respect to comment (1), we have added the structure of meradimate to Figure 1 of the manuscript, by showing the additional menthyl unit in blue.

With respect to comment (2), we have added a table with this information to the manuscript, which is now referred to as Table 1. In lines 62-64, there a reference to this new table has also been added: “The current maximum permissible concentration of avobenzone (and other UV filters) in ‘ready to use’ formulations is given in Table 1.”

With respect to comment (3), we have removed Figure 5 form the manuscript.

With respect to comment (4), we have added the clarification regarding the fact that this work pertains to in vitro tests in the title, abstract and keywords.

We hope these changes are acceptable to the reviewer, making the manuscript fit for publication.

Round 2

Reviewer 1 Report

The authors  have addressed  all the  issues. I  reccomend the paper  for publication.

Author Response

We thank reviewer 1 for a further read of the manuscript and are glad to know they have recommended the paper for publication.